# The Indirect Role of Passive-Avoidant and Transformational Leadership through Job and Team Level Stressors on Workplace Cyberbullying

**DOI:** 10.3390/ijerph192315984

**Published:** 2022-11-30

**Authors:** Jan Philipp Czakert, Rita Berger

**Affiliations:** Department of Social and Quantitative Psychology, Universitat de Barcelona, 08035 Barcelona, Spain

**Keywords:** workplace cyberbullying, passive-avoidant leadership, transformational leadership, work environment hypothesis, job demands–resources model, stress

## Abstract

Research on workplace cyberbullying (WCB) is still scarce and needs verification. This study addressed the indirect influence of positive and negative leadership on WCB via perceived role stressors and negative team climate. The main goal is to test the applicability of the work environment hypothesis and job demands–resources model for WCB on a cross-sectional sample of n = 583 workers in Germany (n = 334) and Spain (n = 249). We tested multiple mediation models, and findings revealed that negative (passive-avoidant) leadership increased role and team stressors and thereby WCB exposure, whereas positive (transformational) leadership decreased the same stressors and thereby reduced WCB exposure. No cross-cultural differences were found, indicating portability of the results. This study highlights the explanatory factors for WCB at individual and team level and emphasizes the role of managers as shapers of the work environmental antecedents of WCB in the emergent digitalized working world. Theoretical implications and future research avenues are discussed.

## 1. Introduction

Workplace bullying (WB) is a complex, worldwide acknowledged problem, and a huge body of occupational research literature has been published on the subject for the past three decades [1]. Representing one of the most severe occupational psychosocial risk factors, WB can cause multiple devastating outcomes, such as burnout, depression, staff turnover and job dissatisfaction; individual and team performance declines and thus affects individual and organizational well-being as well as national economies [1,2,3,4]. Economically, even conservative estimates of financial burden associated with WB declare it as a substantial and sizeable cost factor [5].

With the use of information and communications technology (ICT) accelerated by the COVID-19 pandemic and expected to stay pervasive beyond, ICT-based unacceptable conducts, such as workplace cyberbullying (WCB), are likely to increase along with the omnipresent use of computer-mediated communication [2,6]. Indeed, shown to be a prevalent phenomenon already before the outbreak of COVID-19 [7], a recent study could show that cyberbullying attitudes and cyberbullying perpetration increased during the pandemic [8]. Recent research has postulated that WCB has an even more detrimental impact on mental health than traditional face-to-face bullying, due to its lack of spatial and temporal boundaries and the lack of mental and physical escape options [9]. In simpler words, WCB “can invade the walls of a home which has normally been considered a safe environment” [1] (p. 23).

Although the international labor organization calls for an urgent update of legislative systems to upgrade protection against WCB and ICT-enabled violence and harassment in the world of work [2], research on antecedents of WCB is not only still scarce but also showed mixed results—which will be addressed subsequently—hampering evidence-based solutions on an organizational level. As scholars previously stated, the antecedents of specific WCB have yet to be fully investigated [1,10], and more research must take place to identify the context of WCB [1]. This lack of evidence is even more concerning since WCB has been argued to constitute a distinct, yet closely related phenomenon compared to traditional WB [2,11] so that previous established organizational prevention measures for WB might not sufficiently cover the emergent issue of WCB. On national legislation and organizational policy level, despite increasing awareness regarding risks of violence and harassment posed by the spread of the ICT tools in the world of work, it was stated that the legal system needs to be updated [2].

Notwithstanding studies focusing on prevalence rates [7,12] and definition and conceptualization issues [13,14], not much is known about the subject in general. Most of what is known about general cyberbullying stems from the considerable number of studies among adolescents and younger populations in school settings, leaving a black box regarding how the phenomenon appears in the work environment with its differing power structures and social relations [15].

Therefore, several research gaps appear. Firstly, results on WCB research on a stressful work context are limited at job level, inconclusive at team level [16,17] and, at cross-cultural level, nearly nonexistent although WCB numbers differ across countries. Until now, it remains unclear if WCB—just as WB—seems to be a likely result out of a conflicting work environment at job level (e.g., role stressors) and at team level (e.g., negative team climate), or if perceptions of WCB rather result from its specific characteristics, different to those of WB. For example, the impoverished nature and ambiguous receiver perceptions of computer-mediated communication have also been theorized to play a critical role for the perception of WCB [1] and in shaping social relations at work in general [18]. This would call for more focus on the person-centered hypothesis and related appraisal processes.

Regarding the identified strong linkages between stressful work environments and WB, specific stressors on job level seem of particular relevance [19]. Here, role stress, including role conflict and overload, have been shown to elicit strain reactions associated with WB behavior [19,20], yet until now, not enough research has existed to draw the same robust conclusions for WCB behavior [13].

Regarding the social dimension at team level, stressful team climates and associated conflicts were closely linked to WB [21], but previous research on specific WCB failed to find significant effects [16]. Thus, the link between a negative team climate and perceived WCB exposure has not been sufficiently clarified until now.

Secondly, from a management perspective, more research is warranted to shed light on the role of managers as shapers of the work environmental antecedents of WCB, as the influence of leaders as agents of the organization has been shown for WB [20,22,23], but the role of the leader has found limited attention in WCB research. For example, research has seemingly focused more on leaders’ influence on the WCB-outcome path [24,25] or has included leadership concepts for scale validation purposes [17]. Only a few studies have included leadership as antecedent for perceived exposure to WCB [10,16]. While Gardner et al. (2016) found no significant relationships between leadership and WCB, Tiamboonprasert and Charoensukmongkol [10] did. These mixed findings regarding the role of leadership in the development process of WCB warrant further investigation.

Moreover, a considerable research gap regarding differences across countries for WCB prevails. As stated in a related chapter, little to no cross-cultural research in this area exists [26]. Since research has shown concerns regarding cultural differences in both WB and WCB enactments and perceptions [26], and regarding differing effect sizes when investigating general leadership–work environment outcomes [27], the leadership–WCB mechanism analyzed in this study might also differ across countries. 

### 1.1. Conceptualization of WCB: Similarities and Differences Compared to Traditional WB

WCB is a relatively young yet emerging research topic [2]. The recent systematic review on cyberbullying in the adult population resulted in only 90 peer-reviewed research articles between 2004 and 2016, with most of them focusing on school or university settings [13]. Although occupational research may learn a lot from research in school and adolescent contexts, conceptualization issues for WCB persist, and theories that explain antecedents of WCB remain scarce [1].

WCB differs from other uncivil online behavior, such as cyber-incivility, in the intent to harm and degree of intensity perceived [14]. Conceptually, it has been suggested that WCB is in one part a specific form of general WB, which refers to a repeated and long-lasting exposure to negative acts, whereby the targeted individual has difficulties defending himself or herself [28]. However, WCB seems to be characterized by some distinct and unique features that require particular attention and are closely related to ICT and the way we use it for social interactions, i.e., computer-mediated communication.

Features of ICT, like its spatial and temporary lack of boundaries, facilitate bullying actions from work peers to address victims at home or anywhere, anytime during day and night. This 24/7 possibility of being cyberbullied destroys the “safe haven” [15] (p. 455) of the victims and can thereby even have more severe impacts on personal well-being than traditional WB that is bound to the workplace and work time. This aspect is also labeled as the “intrusive nature” [29] (p. 325) of WCB, as it can intrude into other life domains of the target.

Adding to this, the increased distribution speed of information using ICT facilitates multiplication of the effect of bullying actions by addressing a huge potential audience. Moreover, enduring accessibility and visibility of text, audio, picture or video data for a long period of time enables WCB victims to continuously review them [1], simulating some sort of repetitiveness [12].

Another feature of computer-mediated communication crucial for cyberbullying represents anonymity or perceived anonymity of the perpetrator [8]: The indirect nature of computer-mediated communication results in potential anonymity of the perpetrator, at least in some online bullying acts, such as hijacking the computer or the publication of denigrating pictures or videos. In this way, perpetrators might perceive a lower hindrance of carrying out aggressive actions [8]. However, computer-mediated communication enacted at work usually leaves a “digital footprint” [1] (p. 8), which might keep WCB perpetrators from acting through clearly aggressive messages and might instead provoke more subtle WCB behaviors [1].

Ultimately, most computer-mediated communication is of an impoverished nature in contrast to face-to-face communication [30]. It lacks non-verbal cues of the sender, which causes a vast array of misunderstandings and misinterpretations regarding the content and intent of a message; these might not occur when it is possible to ask for clarification immediately or when talking eye-to-eye [31]. It also lacks the non-verbal cues of the recipient of the message, which leads arguably to lower empathy of the alleged perpetrator [32]. When not receiving clear feedback that communication was perceived in a negative and destructive way, this way of communication might intentionally but also unintentionally proceed [18]. Mounting evidence synergized from management information systems, computer-mediated communication and organizational behavior research suggests that these mentioned features and the pervasive use of computer-mediated communication in the world of work changes relational aspects and that decreased emotional support and increased social undermining, such as WCB, is likely [18].

As a conclusion, it seems that WCB as a concept might also be in core very similar to traditional WB, but its specific features might change enactment and perception and thus warrant a proper investigation [1].

### 1.2. Work-Related Stress as Explanation for WCB

Regarding explanations for WCB, due to the considerable overlap of WB and WCB, many of the explanations found for WB should be applicable for WCB, too [1]. Both WB and WCB have been conceptualized as a behavioral strain to a stressful work environment [17,33]. The claim that stressful work environments serve as a main precursor for WB has found many arguments in occupational research [14,29,34] and is reflected in the work environment hypothesis [35]. The hypothesis claims that bullying is a result of prevailing problems within the psychosocial work environment and not a consequence of individual or interpersonal causal factors [35]. In contrast to this hypothesis stands the person-centered hypothesis, which suggests that individual characteristics play a key role on both the perpetrator and target side [1]. While acknowledging the person-centered hypothesis and its utility from a personnel management (e.g., recruitment or succession planning) perspective, the work environment hypothesis seems more useful when focusing on the prevention of WB from an organizational management and work design perspective [36,37]. Also, meta-analytic findings provide relatively more support for the contextual perspective: While role stressors were found to be uniquely related to WB (accounting together for 21% of the variance in WB), it was stated that personality differences among individuals have only very small influence on perceived WB victimization [20]. Thus, backed by synthetic empirical evidence, the work environment hypothesis amplifies the predominant importance of minimizing stressful psychosocial characteristics of the work environment instead of individualized measures in the prevention of WB and WCB.

Given the strong theoretical linkage to work stress, related occupational stress theories, such as the well-established job demands–resources model [38], have been used to argue for a considerable conceptual overlap between WCB and WB [14]. Notably, both WCB perpetration and perceived WCB exposure have been considered outcomes of stress-related work contexts [29]. In line with WB-research and general occupational stress research [39], previous WCB-research has here distinguished between job-related stressors (i.e., role conflicts); team-related stressors (i.e., interpersonal conflicts in teams, including leadership); and organizational-related stressors (i.e., social climate and organizational change) [29].

Among job-related stressors, role stress has received most attention in WB research and might be also applicable for WCB [1,19]. Two well-known role stress concepts in WB research are role conflict and role overload [20,34]. Role conflict is conceptualized as the perception of the simultaneous existence of two or more sets of expectations towards the same person, such that compliance with one set of expectations conflicts with the compliance of another set of expectations [40]. Role overload refers to the perception of the focal person of having too much work in too little time [41]. In line with the job demands–resources model, both role stressors represent job demands that result in strain in the focal person, for example, in forms of stress, frustration or lowered self-esteem [42,43]. Employees that perceive role conflict may also trigger WCB behaviors directed towards them, as the “victim precipitation theory” [44] suggests: Employees that face high levels of role stress might experience lower confidence in their work-related actions or may act in ways that irritate or annoys their peers, which would provoke incivility behaviors or make it easier to blame them [43]. Little research has shown that role stress is also associated with WCB [17,45].

Thus, we hypothesize that:
**Hypothesis** **1 (H1).***Perceived role stress is positively related to perceived WCB exposure.*

Whereas role stress is rather job-related, team-related stressors represent the social level of the work environment. Empirically, the link between team-related stressors on WB seems well-established [21]. In general, research suggests that, on a more general level, the social organizational climate is also related to WCB [16,24]. On team level, team climate can be referred to as the set of perceptions regarding norms, attitudes and expectations on a team [46]. Psychosocial characteristics, such as team conflict, have been proposed as a precursor [29]. Alike for WB, a dysfunctional team was considered to lead to WCB victimization [47], and the perpetrator source might often be another team member [48]. While a positive team climate might be considered an important resource in the provision of instrumental or emotional support highly needed in bullying processes [16], a negative team climate might deter access to these important resources and, moreover, spur WCB behaviors aimed at retaliation or emotional release [49]. Additionally, one or more members of a work team could repeatedly act as bystanders of WCB, for example, by sharing defamatory pictures or videos through the internet, commenting or liking uncivil posts, etc. (see [1] for more). Yet surprisingly, when contrasting effects of team-related stressors on WB and WCB, research has shown mixed findings. For example, interpersonal conflict was shown to be related to WCB [16], whereas another study showed that team conflict was not significantly related to WCB [16]. To add empirical evidence to these inconclusive findings, we focus on negative team climate as an important team-related stressor for WCB. 

Thus, we further hypothesize that:
**Hypothesis** **2 (H2).***Negative team climate is positively related to perceived WCB exposure.*

Following these considerations, we position perceived WCB exposure as the outcome of perceived job demands on job and team level.

### 1.3. The Role of Leadership on a Stressful Work Environment

By using established leadership theory, leadership has been a well-investigated factor regarding its impact on a stressful work environment. One of the most impactful psychosocial working conditions represents perceived leadership as it shapes the perception of other psychosocial characteristics [22,50]. Following this, leadership has been successfully positioned within the job demands–resources as a standalone predictive factor that affects both job demands and resources [50,51]. Widely regarded as a contextual variable on macro level [52], this indirect influence of leadership is in line with the social environment model [53], which argues that individual perceptions of a leader shape the individual perceptions of experienced work characteristics, which in turn relate to outcomes and strains. Drawing on this and in line with a current chapter on the multiple roles of leaders in the process of WB [54], it seems reasonable to hypothesize that leadership also has a standalone function when it comes to WCB. In this way, we specify the work environment hypothesis hierarchically and put special emphasis on the macro-contextual role of leadership.

The role of leadership has been analyzed also as a precursor of WB over the last years. Indeed, the way direct supervisors treat their staff has been seen to have a direct severe impact on the prevalence of WB [20,22,23]. Regarding varying forms of supervision that can either promote or buffer stressful work environments and thereby either help or hurt the well-being of their subordinates, scholars have thus differentiated between destructive and constructive leadership styles [55,56]. Also, in specific WB research, the categorization between destructive leadership styles that promote or tolerate WB on the one hand and constructive styles which deter harassing actions on the other hand was used to investigate corresponding behavioral patterns of the leader [57,58]. Adopting this perspective, destructive leadership styles, which can be distinguished by active and passive styles [37], can have a direct and indirect influence on WB. As this study focuses on pathways through work stressors, the following paragraphs focus on the indirect influence.

### 1.4. The Indirect Role of Destructive Leadership as Antecedent of WCB

#### 1.4.1. Active Destructive Leadership Styles

Inadequacies and weak leadership skills have been argued to be a main precursor of WB since they contribute significantly to a stressful work environment [43,54]. Representing a stressor in itself, inadequate supervisor behavior was then explained by researchers as an abuse of the power that comes within the higher position, leading to the case when the leadership behavior itself was perceived to contain bullying actions. In this regard, research labeled these leadership styles autocratic leadership or tyranny [37]. Aligned with the social environment model [53], earlier research also argued that active destructive leadership practices have an indirect impact on WB by its considerable effect on the presence of stressors and particularly of role stress within organizations, because conflicting communication from persons of higher hierarchical positions seem to cause more stress than ones from other less powerful role senders [59].

And indeed, supervisors themselves played an active role in many cases of WB: According to a study by leading researchers in the field, 50% of all WB detected cases involved a superior as the accused perpetrator [60]. Regarding WCB, however, the scant findings of recent research could not replicate this role of leadership. In a recent longitudinal study on predictors of WB and WCB in New Zealand, destructive leadership could be linked significantly to traditional WB but not to WCB exposure [16]. Another research work assessed autocratic leadership style for criterion validity purposes of a self-developed questionnaire to measure WCB [17]. In the named study, autocratic leadership was significantly but only very weakly correlated with WCB (0.06, *p* < 0.05), which stands in contrast to WB research. All authors speculate that those not apparent or weak or linkages might be due to the unique characteristics of WCB, but they do not specify in which way [16,17]. Since WCB constitutes, by definition, a more indirect form of bullying [1], active forms of destructive leadership seem thus to play a different role for this phenomenon.

#### 1.4.2. Passive Destructive Leadership Styles

Yet, not only active but also passive behavior of the supervisor has been investigated as antecedent of WB. As research on WCB and passive destructive leadership is lacking, it is referred to findings of WB research.

In this regard, the most extreme form of such weak leadership qualities is the sheer abstention of leadership, called laissez-faire or passive-avoidant, which had been investigated by numerous researchers as a precursor of bullying [4,22,61]. Passive-avoidant leadership represents both the passive mode of reaction and the lack of reaction at all [62], combining the dimension of management by exception passive and laissez-faire style of the full range leadership model [63]. Notably, a study by Tsuno and Kawakami [23] found that employees working under higher laissez-faire leadership had 4.3 times higher risk of new bullying cases than the comparison group. Scientists thus argue that passive-avoidant leadership is not a type of zero-leadership but more a type of destructive leadership [51].

As research suggests, passive-avoidant leadership seems to create a stressful environment in which rules and social norms are no longer obeyed [64] and abusive social behaviors are tolerated or not sanctioned [57].

Following our theory development based on the job demands–resources, we thus conceptualize passive-avoidant leadership as a job demand that increases the effect of work stressors on both job-related and team-related levels, and thereby the likelihood of WCB.

Regarding role stress, passive-avoidant leadership has shown to increase role conflict and role overload [64,65]. Expectation management and sharing necessary information regarding successful task completion is a basic leadership function, and failure of this function results in conditions of perceived overload and conflict [64]. Perceived overload and conflict are likely connected to the missing guidance related to passive-avoidant leadership, as direction for task prioritization is lacking and resource adaptation resulting out of additional task input is not managed effectively [64].

Additionally, regarding negative team climate, research could show that passive-avoidant leadership adversely influences the team climate [51,56]. For example, it was suggested that the lack of constructive leadership is linked to considerable levels of frustration among the employees [56], which could trigger a negative team climate. Moreover, the lack of proper corrective leadership actions for interpersonal conflict processes might result in a negative team climate [65]. Specifically, passive-avoidant leadership might signal not only to tolerate WB actions but also to lower the risks on the side of the bully to be confronted with negative consequences, which facilitates peer-to-peer bullying [22,54,61]. This might be particularly relevant for WCB, as research suggests that sources of WCB are more frequently peers than supervisors [48].

Given this, we hypothesize the following:
**Hypothesis** **3 (H3).***Passive-avoidant leadership has a positive indirect effect of perceived WCB through perceived increased role stress and perceived increased negative team climate.*

### 1.5. The Indirect Role of Constructive Leadership as Antecedent of WCB

Turning the eye from the problem-oriented towards the solution-oriented view, constructive leadership—which is not separated into active or passive categories as constructive leadership is active leadership [66]—is claimed to deter WB actions [58]. Constructive leadership styles that have been most investigated as antecedent for WB represent transformational leadership [66,67,68] and ethical leadership [69], whereas the latter was investigated also for WCB [10,16]. However, the utility of investigating ethical leadership instead of transformational leadership has been questioned, since ethical and moral behavior has been shown to represent a central component of transformational leadership [70]. Moreover, the issue of construct proliferation of positive leadership styles has been raised, which hampers parsimony [71]. Given this, and since the link between transformational leadership and work-related job demands is well-established [55], we focus on transformational leadership in this study.

There is common agreement in research that transformational leadership has a positive influence on the prevention of WB actions. Here again following the social environment model [53], it was argued that supervisors portraying transformational leadership qualities contribute to a healthy social environment by promoting pro-social values [68] and more justice [66]. Regarding role stress, by addressing and respecting individual needs and offering support, psychological impacts of stressors that could potentially lead to WB behaviors, such as frustration and anger, can be reduced [67]. Contrasting passive-avoidant leadership, transformational leadership represents effective direction, feedback and guidance that safeguard the employees and reduce potential role stress related to overload and conflict [64].

Concerning leadership and its link to team climate, it is generally found that leaders have the power to shape the social climate significantly [42]. Regarding transformational leadership and negative team climate, it was suggested that employees who find themselves in interpersonal conflicts not escalated yet can address the issue with their supervisor, facilitating quick conflict interventions. According to scholars, transformational leaders do not neglect nor ignore conflicts but rather see them as a challenge that can be solved only by collaboration, which enhances the likelihood of effective team conflict management [66] (see also [42]). The direct contributions to a positive social climate buffer the relationships between other work-related stressors and WB behaviors, contrary to passive-avoidant leadership. Recent evidence supports this indirect influence of leadership on WCB exposure through its positive impact on the team climate [10].

Following our argumentation, we hypothesize:
**Hypothesis** **4 (H4).***Transformational leadership has a negative indirect effect of perceived WCB through perceived reduced role stress and perceived reduced negative team climate.*

### 1.6. Leadership—WCB Relationship in Germany and Spain

Cultural differences across countries regarding WCB perceptions and enactments are under-researched [26]. This is concerning because of three main aspects that warrant an integration of the cross-cultural perspective: Firstly, research on traditional WB has already shown that national cultures influence WB [37]. Secondly, computer-mediated communication behavior relevant for WCB also varies across countries [26]. Thirdly, general differences prevail when investigating leadership—work environment outcomes [27], thus the leadership—WCB mechanism studied in this manuscript might also differ across countries. Given these considerations, and to increase the portability of our findings, we build our study on a sample from Germany and Spain.

When looking at prevalence differences of WCB between Germany and Spain, previous research could show that the WCB mean was higher in Spain than in Germany [45], but the spread of ICT-based working is similar [72]. It was argued that this difference in perceived WCB exposure might be explained by the cultural differences in power distance (Spain = 57, Germany = 35), and the rather broad assumption that cultures scoring higher on power distance might be more prone to WCB [37]. In this study, we hypothesized that the dimension of power distance also explains higher effect sizes for the leadership-stressors relationship. In line with the GLOBE studies [73], it was assumed that, in cultures with higher power distance, the effect of leadership on perceived work environment outcomes might generally be higher [27]. In simpler words, it was assumed that the Spanish working culture accepts greater power disparities and thus allows the leader more power over work environment features.

**Hypothesis** **5 (H5).**
*The effect sizes of leadership on role stress and negative team climate are higher in Spain than in Germany.*


### 1.7. The Present Study

Given the mixed and limited context-related and cross-cultural research, this study aimed to contribute to the research fields as follows. Firstly, this study uses environment-related models building on the prominent work environment hypothesis [35] as well as on established stressor/strain theories in the form of an integrative job demands–resources model [38,42] to test with role stress a perceived job-related demand and with negative team climate a team-related demand and to clarify the role of stressors on different levels for WCB. We thereby contribute to the limited research that used the job demands–resources for WCB. Secondly, by including both destructive and constructive leadership styles as contrasting antecedents of these perceived stressors at different levels and ultimately of perceived WCB exposure, we add empirical evidence and a differential view to the scarce leadership–WCB research. In doing so, we emphasize the important indirect influence of leadership on WCB. This is reflected in the social environment model [53] and the well-established idea that leadership plays a key role in managing the allocation and impact of psychosocial characteristics at work [50]. Specifically, we tested multiple mediation models, including transformational and its opposite passive-avoidant leadership, role stress and negative team climate as mediators, and perceived WCB exposure as outcome. Thirdly, we contributed to the scarce cross-cultural research in this area. Since prevalence rates have been shown to diverge among Germany and Spain [45] while the spread of ICT-based working is quite similar [72], we compared our findings between these two culturally different countries to increase the portability of our findings.

Summarized, Figure 1 and Figure 2 illustrate our hypothesized models (one passive-avoidant leadership model, one transformational leadership model).

## 2. Materials and Methods

### 2.1. Study Design and Sampling

This study was part of a pilot study of the Erasmus+-project “Improving management competences on Excellence based Stress avoidance and working towards sustainable organizational development in Europe” (IMPRESS), grant number 588315-EPP-1-2017-1-ES-EPPKA2-KA, and was carried out in accordance with the recommendations of the Ethical Principles of Psychologists and Code of Conduct by the American Psychological Association [74] and in accordance with the Declaration of Helsinki [75]. The survey, named “Stress at work—survey”, was rolled out as a public web-based survey. Data analysis was conducted via IBM SPSS Statistics 25 (IBM corp., Armonk, NY, USA). 

### 2.2. Participant Recruitment

Snowballing procedure was applied in Germany and Spain across all possible occupational sectors and regardless of company size. Since the survey-link was open and promoted via social media, no response rate was assessed. Several channels were used to promote the questionnaire in Germany. The open survey-link was distributed within universities and businesses of various branches covering small producing companies, consulting companies, training institutions, scientists, trade fair organizers and also non-governmental organizations. In Spain, administrative personnel of the University of Barcelona as well as master’s students that were working were encouraged to distribute the survey through their professional networks. Other partners of the research project sent the survey to employees and personal contacts of various business organizations located in Spain. The participants were given a comprehensive consent form including detailed information about the project and the anonymization of the data. The survey addressed people working part-time or fulltime. All participants were notified about the voluntary nature of the study. 

### 2.3. German Sample Characteristics

In Germany, data were retrieved from 334 participants. Among them, 200 (60.2 percent) were female, and 132 (39.8 percent) were male. Overall mean age was M = 33.46 years (SD = 12.43; range 18–70). Regarding participants’ job position, 282 (85.5 percent) were currently subordinates, and 48 (14.5 percent) were supervisors.

### 2.4. Spanish Sample Characteristics

A total of 249 participants completed the questionnaire. Among them, 141 (56.6 percent) were female, and 87 (34.9 percent) were male; one participant selected the gender category “other”. The overall mean age was M = 43.07 years (SD = 9.73; range 18–64). One hundred eighty-eight (75.5 percent) respondents were currently subordinates, and 40 (16.1 percent) were supervisors.

### 2.5. Measures

We followed the guidelines of the International Test Commission (2017) [76], including forward and backward translation procedures by native Spaniards and Germans who were also experts in the field of occupational psychology research. For the CBQ-S, measurement invariance was already rigorously tested for Spain and Germany [45]. The items were introduced with the following question: “When I think about my work, to what degree do these aspects cause me stress?”. Participants could then respond on a response scale ranging from 1 = Aspect does not exist; 2 = Causes not at all stress; 3 = Causes very little stress; 4 = Causes to some degree stress; to 5 = Causes to a very great degree stress.

Perceived WCB exposure was measured by the validated six-item version of the CBQ–S [77] for Spain and Germany [45]. The CBQ–S is a behavioral inventory or behavioral experience method, typically used in bullying research [28]. An example item is “being threatened online”. Cronbach’s alpha was 0.92 for the German sample and 0.94 for the Spanish sample. 

Role stress was measured by four items, including items regarding role conflict and role overload [41] (see also [40]). An example item for role conflict is “receiving incompatible requests” and, for role overload, “receiving an assignment without the manpower to complete it” [41]. Cronbach’s alpha was 0.83 for the German sample and 0.85 for the Spanish sample. 

Negative team climate was measured by three items based on the Job Stress Survey [78]. An example item is “colleagues not doing their job”. Cronbach’s alpha was 0.82 for the German sample and 0.87 for the Spanish sample.

Passive-avoidant leadership was assessed by using four items of the Spanish and German versions of the well-established Multifactor Leadership Questionnaire (MLQ–5X) developed by Avolio and Bass (2004) [79]. An example item for Laissez-faire leadership was “supervisor who tries to keep out of critical matters” and, for management by exception passive, “supervisor who reacts only after errors have occurred”. Cronbach’s alpha was 0.92 for the German sample and 0.95 for the Spanish sample.

Transformational leadership was measured using the eight-item Spanish [80] and German version [81] of the Human System Audit–TFL short-scale (HSA–TFL–ES and HSA–TFL–DE), developed by Berger et al. [80] and as part of the Human System Audit framework for human resources quality assessment [82]. An example item of the HSA–TFL–ES (HSA–TFL–DE, respectively) is “I have trust in my supervisor’s ability to overcome any obstacle”. Cronbach’s alpha was 0.95 for the German sample and 0.98 for the Spanish sample. 

In the present questionnaire, the transformational leadership-related items were introduced with the following question: “When I think about my work, to what degree do these aspects cause me relief?”. Participants could then respond on a response scale ranging from 1 = not at all; 2 = very little; 3 = to some degree; 4 = to a great degree; 5 = to a very great degree.

### 2.6. Statistical Analyses

Due to potential common method variance, Harman’s single-factor test was applied to check if one single factor accounts for most of the covariance (i.e., more than 50%) of the variables by using unrotated principal axis factoring in SPSS [83]. In both samples, no single general factor emerged, hence it was assumed that common method bias was not a considerable issue for the present data. 

The CBQ–S variables were log-transformed prior to analysis, since they showed the expected right-skewed distribution, common to bullying scales [16]. Pearson correlations were calculated to analyze the relationship between the selected constructs. The PROCESS extension module was used for mediation analyses since it allows the conducting of multiple simple regression analyses to run by only issuing one command [84]. To further overcome potential issues of nonnormality, 5.000 bootstrap samples with 95%-Confidence Intervals (CI) were applied [84].

## 3. Results

The descriptive statistics as well as Pearson correlations between leadership, job demands and WCB are shown in Table 1 for the German (n = 334) and Spanish (n = 249) sample. 

H1 and H2 were fully supported in both samples. Regarding H1, correlation coefficients between role stress and WCB were r = 0.44 (German sample) and r = 0.65 (Spanish sample). As for H2, negative team climate was positively correlated to WCB in both samples, with r = 0.53 (German sample) and r = 0.72 (Spanish sample), correspondingly. 

H3 was also fully supported. Passive-avoidant leadership was positively correlated with role stress, with r = 0.57 (German sample) and r = 0.65 (Spanish sample), correspondingly. Passive-avoidant leadership was also positively correlated to negative team climate, with r = 0.67 (German sample) and r = 0.70 (Spanish sample), respectively. The multiple mediated regression model showed significant indirect effects of passive-avoidant leadership on WCB via role stress (95% CI [0.010, 0.045]) and via negative team climate (95% CI [0.039, 0.086]) for the German sample (*p*-value < 0.05). Likewise, regression results were significant for the Spanish sample for both paths: passive-avoidant leadership on WCB via role stress (95% CI [0.010, 0.050]) and via negative team climate (95% CI [0.020, 0.069]). The significant paths (*p*-value < 0.05), including the unstandardized regression coefficients for the two samples, are visualized in Figure 3 and Figure 4.

H4 was fully supported in both samples. Negative Pearson correlation coefficient values for the link between transformational leadership and role stress were r = −0.17 (German sample) and r = −0.23 (Spanish sample), correspondingly. Transformational leadership was also negatively correlated with negative team climate, with r = −0.16 (German sample) and r = −0.20 (Spanish sample), respectively. The multiple mediated regression model showed significant indirect effects of transformational leadership on WCB via role stress (95% CI [−0.010, −0.001]) and via negative team climate (95% CI [−0.016, −0.002]) for the German sample. Likewise, regression results were significant for the Spanish sample for both paths: transformational leadership on WCB via role stress (95% CI [−0.021, −0.005]) and via negative team climate (95% CI [−0.025, −0.004]). The significant paths (*p*-value < 0.05), including the unstandardized regression coefficients for the two samples, are visualized in Figure 5 and Figure 6.

H5 was not supported, as the effect sizes of leadership on both mediators were not significantly different between Germany and Spain.

## 4. Discussion

The main goal of the study was to test the work environment hypothesis and job demands–resources model for WCB using role stress and negative team climate and to investigate the indirect role of perceived leadership as antecedent of perceived WCB exposure. Particularly, the objective was to link leadership to the perception of work stress on different levels as antecedents for perceived WCB exposure. A secondary goal was to contribute to the scarce cross-cultural research in this area. To achieve these goals, the potential indirect effects of two contrasting leadership styles’ passive-avoidant leadership and transformational leadership on perceived WCB exposure through role stress and negative team climate was tested in Germany and Spain. 

As H1–H4 were fully supported, the results of the study generally endorse the ideas proposed by the work environment hypothesis and social work environment model. Leadership as contextual macro-resource—or macro-demand—shapes the way employees perceive major stressors. As hypothesized, the way employees perceive constructive leadership or destructive leadership has an impact on the odds of perceived exposure to WCB, through its impact on the perception of job-related and team-related stressors. 

Regarding H3 and the significant impact of passive-avoidant leadership on WCB through role stress and negative team climate, the results suggest that such passive forms of leadership increase the odds of WCB exposure by shaping the perception of role stress related to role conflict and role overload. Indeed, passive-avoidant leadership amplified the perception of role stress, corroborating previous research findings [51,64]. Like results regarding perceived WB exposure [34], perceived incompatible requests and perceived lack of resources to cope with the amount of workload might lead to perceived WCB exposure. Also, in line with a previous meta-analysis on WB [20], the found correlation coefficients between role stress and WCB suggest that this relationship has similar intensity when it comes to WCB. Yet, as the WCB data had been log-transformed prior to regressions analyses, we refrain from interpreting the intensity of the unstandardized regression coefficients.

Passive-avoidant leadership also increased the team-related stressor negative team climate in both samples significantly, which was linked to increased WCB exposure. This supports previous research that highlighted the adverse effects of lack of constructive leadership on different components of a team climate (e.g., team conflict management climate [42]; team climate for learning [51], see also [56]). The high correlations between negative team climate and WCB suggest that the concepts are closely linked with each other. This is in line with previous findings on WCB, which declare WCB as a mainly peer-to-peer form of WB [48]. Nevertheless, the mediation analysis results could show that passive-avoidant leadership significantly impacts the perception of negative team climate and thereby contributes to team-related antecedents of WCB. 

Summarized, regarding passive-avoidant leadership this study highlights that, once again, passive-avoidant leadership is not a zero-sum leadership style but a destructive one. Moreover, this research could clarify the important role of passive forms of destructive leadership for WCB as influencers of a stressful work environment. Whereas in WB, active forms of destructive leadership (the leader as a bully) are making a large share of WB cases [60], active forms were not or were only weakly linked to WCB [16,17]. One explanation for this might lie in the traceability of computer-mediated communication, which would make active aggressive behavior less likely [1]. Hence, as subtle forms of WCB might increase [1], so does the indirect role of leadership as work stress manager. In line with Vranjes et al. [17], which proposed similar stress/strain pathways for both WCB exposure and perpetration, the absence of leadership increases other work-related stressors, and thus an environment where perceived WCB exposure is likely to happen. When management is poor, cyberbullies might not fear negative consequences like sanctions and thus might proceed with the subtle aggressive online attacks [37]. 

Transformational leadership had significant indirect effects on perceived WCB exposure through decreased perception of job-related (role stress) and team-related (negative team climate) stressors in both samples and countries.

Regarding H4 and alike previous WB research, it could be shown that transformational leadership as a contextual macro resource can decrease role stress that is linked to bullying behavior [58]. In other words, perceived transformational leadership decreased the feeling of receiving conflicting tasks and expectations from peers as well as overload and thereby decreased the perception of WCB. In terms of the transformational leadership–role stress relationship, it must be stated that, although significant, correlation estimates were rather low compared to the passive-avoidant leadership–role stress relationship, with r = −0.15 (Spanish sample) and r = −0.17 (German sample), correspondingly. This corroborates previous research that contrasted transformational leadership and passive-avoidant leadership and found that relationships between passive-avoidant leadership and job demands are in general stronger than the negative relationship between transformational leadership and job demands [51]. 

Also, in line with H4, transformational leadership decreased negative team climate and thereby perceived WCB exposure in both countries. This corroborates previous findings of ethical leadership and WCB [10], who could show a similar mechanism in Thailand. Also, it is in line with transformational leadership theory, which argues that transformational leaders tackle interpersonal conflicts from early on and excel generally in conflict management [49], and significantly influence the conflict management ability of teams [42]. Additionally, behavior associated with transformational leadership seems to lower the psychosocially negative outcomes of work-related stressors [49].

### 4.1. The Cross-Cultural Perspective on the Leadership–WCB Relationship

Contrary to transformational leadership research [27], WB research [37] and prevalence research on WCB [45], culture did not play a role in the investigated mechanism. Therefore, we suggest that leadership, as perceived effective or ineffective work stress manager, plays a vital role in the perception of WCB exposure in Spain and in Germany. 

### 4.2. Limitations and Future Research

Since the study followed a cross-sectional design, statistical measures were limited. Although we use terms such as “effect” throughout the manuscript for ease of reading, we put strong emphasis to highest caution on inferring causal pathways of the present study results. However, other authors and experts in the field of mediation analyses pointed out that mediation is a causal phenomenon that relies mainly on strong theoretical arguments [84], which is why we built on established theories. 

The second limitation concerned the general data collection. That is, it only used one source and method, namely self-reports from working persons. This raises concerns of common method bias [83]. In organizational research, common method variance is estimated to cause about 26% bias in the observed relationships of constructs [85]. Although we applied procedural remedies and Harman’s single factor test to reduce and check for potential common method variance, we recommend testing the results using a longitudinal research design.

Also, we emphasize that the reliance on self-reported data might conflate behavioral concepts with nonbehavioral concepts: Perceptions (i.e., I feel exposed to WCB) and evaluations (i.e., I evaluate my direct supervisor as transformational) are only proxies of behavior, which is associated with a range of biases and hampers theory testing [86]. Notably, this is a general issue in organizational behavior research and especially in leadership–follower studies [86]. We suggest that this might be of particular concern in WCB research, as the unique features of computer-mediated communication strongly emphasize the role of perceptions of intentions and leave a big space for interpretations [31]. Therefore, to advance the field, it would be highly advantageous for future studies to use multiple sources of information (e.g., coworkers, direct supervisors, occupational health reports) and multiple methods, including behavioral concepts.

The third limitation addresses the sample. Although only workers were addressed, and we applied a wide range of distribution channels to enhance representativity and heterogeneity, our samples for Germany and Spain were only of moderate size and thus may not have been representative of the entire Spanish and German employed population to which we would like to generalize. However, the size was acceptable for the conducted statistical measures, and the applied bootstrapping procedures and resulting confidence intervals often lead to better estimates of true effect sizes than any single sample might [87]. Future research should use probabilistic sampling techniques and control for the occupational sector of the participants to distinguish appropriately between specific or generalizable results. Also, local companies compared to multinational companies might display cultural differences better. However, we localized all measurements in the respective national language, Spanish and German. 

The fourth limitation lies in the distinction between WCB and other related concepts. Due to the apparent blurred lines between WCB and other similar concepts, such as cyberincivility [14], discriminant validity should be tested to increase construct validity. The fact that intent to harm seems to be a distinctive aspect for these two concepts [14], but at the same time is widely excluded from WCB definitions [1], warrants more conceptual clarifications in future research.

Notwithstanding these limitations, it is believed that the current study has contributed substantially to the scientific body of WCB. By showing the importance of destructive and constructive leadership in the development stage of WCB, this research has provided more arguments against the absence of leadership and for promoting and engaging in transformational leadership behavior to reduce a stressful work environment. 

### 4.3. Theoretical and Practical Implications

Theoretically, these findings imply that managers’ behavior be brought into focus when investigating the sources of WCB. The integration of the social environment model [53], the job demands–resources model for leadership [32] and the work environment hypothesis for WB [35] provide a fruitful theoretical basis to discuss impacts and levers of leadership as prevention measure for WCB. The prominent stand-alone function of perceived leadership organizes the broader work environment hypothesis hierarchically and thereby specifies WCB prevention measures on different levels (leadership, job-related, team-related).

Practically, the findings imply that the perception of the leader as stress manager [55] shapes impactfully the perception of job demands and resources. From a negative perspective, practitioners should deter passive-avoidant leadership as it at least does not tackle work-related stressors appropriately. This in turn opens doors for emotional-driven reactions, such as WCB, as a result to the stressful experience at work.

From a positive perspective, one possible way to deter WCB may lie in fostering transformational leadership, as the study results suggest. However, the relatively low coefficient values question the sufficiency of the promotion of transformational leadership when taking preventive actions for WCB. Alike most (if not all) social phenomena, WCB is also multidetermined, meaning that multiple factors play an antecedent role here [1]. Additionally, research has shown that WB preventive actions from the leader have been shown to be perceived most effective when they were consistent with organizational policies [57]. However, organizational policies that regulate computer-mediated communication and discuss WCB are widely lacking [2], which constitutes boundary conditions for the leadership–WCB lever. This said, prevention measures for WCB should combine multiple actions on multiple levels (e.g., on organizational policy level, on leadership level, on work design level for role stress; on interpersonal level for negative team climate).

## 5. Conclusions

WCB is a worldwide phenomenon and an increasing psychosocial risk factor in our computer-mediated communication-based modern workplaces. Most of our knowledge on the issue comes from adolescent and youth research and general WB literature, and mixed findings regarding WCB antecedents warrant more investigation. Being an emerging topic in occupational health research, this study integrated the topic into occupational stress theory and could show cross-culturally that a psychosocially stressful work environment in the form of job-related and team-related stressors facilitates WCB in a similar way as for WB. Contradictory to the traditional view of WB as a mainly top-down process, this new form of bullying at work might arguably disregard hierarchies and rather takes place on the horizontal level. This, and the fact that online communication is difficult to supervise, made the role of constructive and destructive leadership questionable for WCB. The present study provided strong arguments that basic leadership functions to reduce work stress are still crucial to deter WCB exposure in the emergent digitalized working world.

## Figures and Tables

**Figure 1 ijerph-19-15984-f001:**
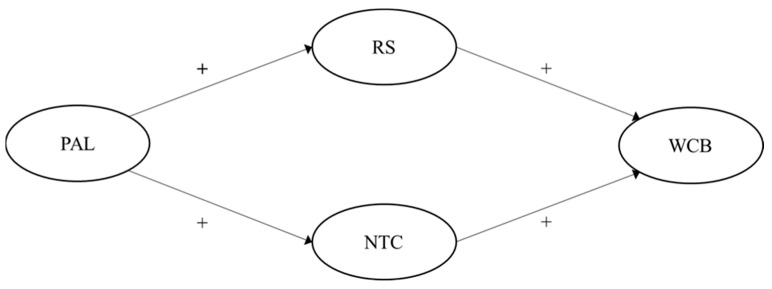
Hypothesized multiple mediation model for passive-avoidant leadership on perceived WCB through role stress and negative team climate. Note: PAL = Passive-avoidant leadership; RS = Role stress; NTC = Negative team climate; WCB = Workplace cyberbullying.

**Figure 2 ijerph-19-15984-f002:**
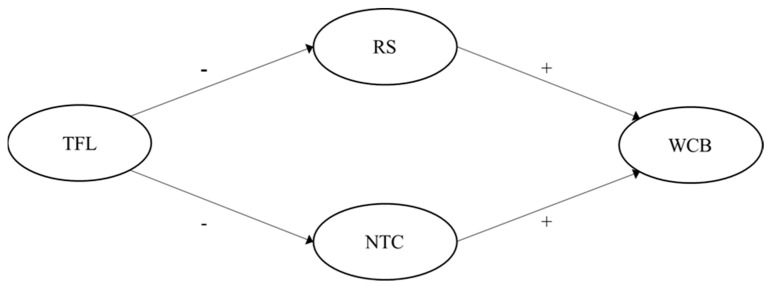
Hypothesized multiple mediation model for transformational leadership on perceived WCB through role stress and negative team climate. Note: TFL = Transformational leadership; RS = Role stress; NTC = Negative team climate; WCB = Workplace cyberbullying.

**Figure 3 ijerph-19-15984-f003:**
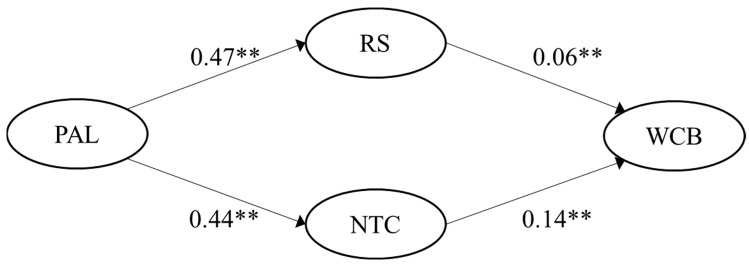
Empirical multiple mediation model for passive-avoidant leadership for the German sample (n = 334). Note: ** indicates a *p*-value < 0.01; PAL = Passive-avoidant leadership; RS = Role stress; NTC = Negative team climate; WCB = Workplace cyberbullying.

**Figure 4 ijerph-19-15984-f004:**
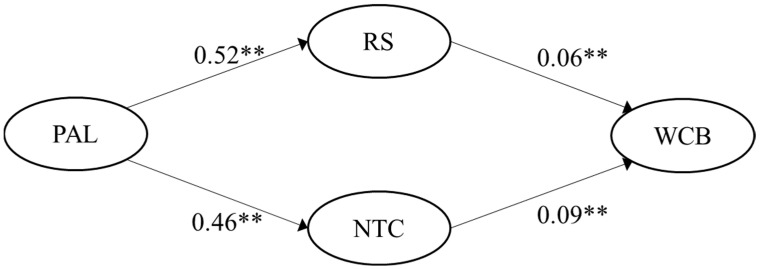
Empirical multiple mediation model for passive-avoidant leadership for the Spanish sample (n = 249). Note: ** indicates a *p*-value < 0.01; PAL = Passive-avoidant leadership; RS = Role stress; NTC = Negative team climate; WCB = Workplace cyberbullying.

**Figure 5 ijerph-19-15984-f005:**
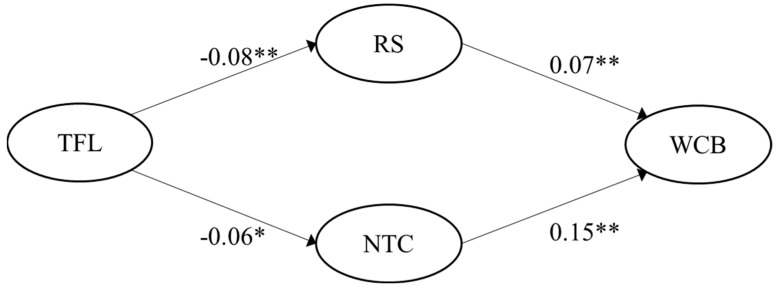
Empirical multiple mediation model for transformational leadership for the German sample (n = 334). Note: ** indicates a *p*-value < 0.01; * indicates a *p*-value < 0.05; TFL = Transformational leadership; RS = Role stress; NTC = Negative team climate; WCB = Workplace cyberbullying.

**Figure 6 ijerph-19-15984-f006:**
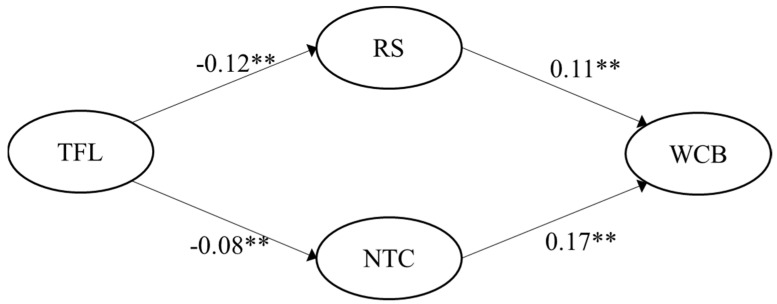
Empirical multiple mediation model for transformational leadership for the Spanish sample (n = 249). Note: ** indicates a *p*-value < 0.01; TFL = Transformational leadership; RS = Role stress; NTC = Negative team climate; WCB = Workplace cyberbullying.

**Table 1 ijerph-19-15984-t001:** Means, standard deviations and Pearson correlations between leadership, job demands and WCB for the German (n = 334) and Spanish (n = 249) sample.

Measure	*M*	*SD*	TFL	PAL	RS	NTC	WCB
TFL	24.65	8.81	1	−0.33 **	−0.23 **	−0.20 **	−0.11
PAL	9.80	5.06	−0.40 **	1	0.65 **	0.70 **	0.72 **
RS	4.71	2.17	−0.17 **	0.57 **	1	0.54 **	0.65 **
NTC	7.47	3.32	−0.16 **	0.67 **	0.57 **	1	0.72 **
WCB	10.45 ^†^	6.35 ^†^	0.05	0.41 **	0.44 **	0.53 **	1

Note. ^†^ Mean and standard deviation prior to log-transformation. ** Correlation is significant at the 0.01 level (2-tailed). Correlation coefficients above the diagonal refer to the Spanish sample; below the diagonal refer to the German sample. TFL = Transformational leadership; PAL = Passive-avoidant leadership; RS = Role stress; NTC = Negative team climate; WCB = Workplace cyberbullying.

## Data Availability

The data presented in this study are available on request from the corresponding author. The data are not publicly available due to commercialization of research findings.

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
