# Peer review of "The Indirect Role of Passive-Avoidant and Transformational Leadership through Job and Team Level Stressors on Workplace Cyberbullying"

_ijerph, 2022, doi:10.3390/ijerph192315984_

Round 1
Reviewer 1 Report
Dear authors,
Thank you for tackling an important topic.
However, I'd suggest you some improvements to be considered:
1) Avoid an excessive use of abbreviations - it deficults the fluidness of reading. Readers should keep in mind all abbreviaitons... If it is not a question of space, I'd leave to main WB & WCB in the text. In the Fugures you may use all, but put a Legend for Abbreviations below.
If you leave more than 2 abbreviations, at least alternate use of full vs. abbreviated terminology, but would descourage to use abbreviations if they are not necessary, especially for Ls, RS, NTC, etc..
2) To the point of the results extrapolation on cultural differences -> a part of Limitaion of a small sample size, also it is a question - I have not seen - where enteprises from where you've got participants were multinational, unternational o local companies. Culture would be seen much better within local, small or family like organizagitons, not in multinational. If you used English, not Spanish, also could reflect on it.
3) About correlations results report - I'd suggest to substitute "ranging" since it might be confusing and readers can perceive about higher (absolute) correlation values in Spain (due to "ranging"), but with a smaller sample (in Spain) an absolute value of correlation can be higher. If the samples were of the same size, than we cold compare, but in this case it is not meaning that in Spain they are greater or smaller. Just report "correlations were .... in Germany and Spain correspondingly".
Author Response
Many thanks for your improvement suggestions. Below you will find the specific answers to your suggestions:
1) Avoid an excessive use of abbreviations - it deficults the fluidness of reading. Readers should keep in mind all abbreviaitons... If it is not a question of space, I'd leave to main WB & WCB in the text. In the Fugures you may use all but put a Legend for Abbreviations below. If you leave more than 2 abbreviations, at least alternate use of full vs. abbreviated terminology, but would descourage to use abbreviations if they are not necessary, especially for Ls, RS, NTC, etc..
Response:
Thank you for this suggestion. We eliminated all abbreviations except WB and WCB, as well as the widely accepted abbreviation ICT for “information and communications technology”. As suggested, we added legends in the notes for figures and the table. We hope this enhances readability of the paper.
2) To the point of the results extrapolation on cultural differences -> a part of Limitaion of a small sample size, also it is a question - I have not seen - where enteprises from where you've got participants were multinational, unternational o local companies. Culture would be seen much better within local, small or family like organizagitons, not in multinational. If you used English, notSpanish, also could reflect on it.
Response:
Thank you for this reflection suggestion. We added information regarding sector under Participants Recruitment in line 507: “…across all possible occupational sectors and regardless of company size”.
We reflected on this aspect and the aspect of company type (local vs. international) under Limitations in lines 758ff: “The third limitation addresses the sample. Although only workers were addressed and we applied a wide range of distribution channels to enhance representativity and heterogeneity, our samples for Germany and Spain were only of moderate size and thus may not have been representative of the entire Spanish and German employed population to which we would like to generalize. However, the size was acceptable for the conducted statistical measures, and the applied bootstrapping procedures and resulting confidence intervals often lead to better estimates of true effect sizes than any single sample might [87]. Future research should use probabilistic sampling techniques and control for the occupational sector of the participants to distinguish appropriately between specific or generalizable results. Also, local companies compared to multinational companies might display cultural differences better. However, we localized all measurements in the respective national language, Spanish and German“. We also add your comment regarding company type for cross-cultural comparisons and emphasize our use of localized scale versions (German and Spanish versions).
3) About correlations results report - I'd suggest to substitute "ranging" since it might be confusing and readers can perceive about higher (absolute) correlation values in Spain (due to "ranging"), but with a smaller sample (in Spain) an absolute value of correlation can be higher. If the samples were of the same size, than we cold compare, but in this case it is not meaning that in Spain they are greater or smaller. Just report "correlations were .... in Germany and Spain correspondingly".
Response:
Thank you for this comment. We substituted “ranging” for our correlational results accordingly in the Resultssection.
Reviewer 2 Report
The indirect role of passive-avoidant and transformational leadership through job and team level stressors on workplace cyberbullying
The idea of ​​the study seems interesting, different and even necessary
Abstract
Is fine followed the solid scientific work structure, but The abstract is long, I suggest that it be restructured and shortened to be more clear and attractive.
Introduction and Literature Review
State the objective at the end of the introduction, as well as the method used and the study's novel contributions.
Methodology.
Fine, but need some answers.
1. How did the authors contact the respondents (the survey-link was open and promoted via social media)?
2. What is the response rate? Does the authors' sample represent the population? Make sure whether the authors' sample can represent the population.
Results.
No comments
Discussions
fine
The conclusions
fine
Others:
The paper has some editing issues.
Good luck
Author Response
The idea of the study seems interesting, different and even necessary.
Response: Thank you.
Many thanks for your improvement suggestions. Below you will find the specific answers to your suggestions.
1) Abstract
Is fine followed the solid scientific work structure, but The abstract is long, I suggest that it be restructured and shortened to be more clear and attractive.
Response:
Abstract: Thank you very much for this comment. We restructured and shortened the abstract to make it clearer and more attractive (lines 11-22): “Research on WCB is still scarce and needs verification. This study addressed the indirect influence of positive and negative leadership on workplace cyberbullying (WCB) via perceived role stressors and negative team climate. The main goal is to test the applicability of the work envi-ronment hypothesis and job demands-resources model for WCB on a cross-sectional sample of N = 583 workers in Germany (n = 334) and Spain (n = 249). We tested multiple mediation models and findings revealed that negative (passive-avoidant) leadership increased role and team stressors and thereby WCB exposure, whereas positive (transformational) leadership decreased the same stressors and thereby reduced WCB exposure. No cross-cultural differences were found, indicating portability of the results. This study highlights the explanatory factors for WCB at individual and team level and emphasizes the role of managers as shapers of the work environmental antecedents of WCB in the emergent digitalized working world. Theoretical implications and future research avenues are discussed.”
2) Introduction and Literature Review
State the objective at the end of the introduction, as well as the method used and the study's novel contributions.
Response:
Introduction: Thank you for this suggestion. We placed the objective, method used, and novel contributions paragraph now at the end of the introduction, labelled “1.7. The present study” in the lines 463-485.
3) Methodology:
Fine, but need some answers.
3.1. How did the authors contact the respondents (the survey-link was open and promoted via social media)?
Response:
Thank you for your comment. We added respective information in the section “2.2. Participant Recruitment” in lines 506-520.
3.2.) What is the response rate? Does the authors' sample represent the population? Make sure whether the authors' sample can represent the population.
Response:
Unfortunately, we could not calculate a response rate (rates of surveys received per contact attempt), as we applied snowballing sampling and had no information regarding the specific number of contact attempts. We now address representativeness of our sample in the limitations section including an additional reference for this in lines 758-769: “The third limitation addresses the sample. Although only workers were addressed and we applied a wide range of distribution channels to enhance representativity and heterogeneity, our samples for Germany and Spain were only of moderate size and thus may not have been representative of the entire Spanish and German employed population to which we would like to generalize. However, the size was acceptable for the conducted statistical measures, and the applied bootstrapping procedures and resulting confidence intervals often lead to better estimates of true effect sizes than any single sample might [87]. Future research should use probabilistic sampling techniques and control for the oc-cupational sector of the participants to distinguish appropriately between specific or gen-eralizable results. Also, local companies compared to multinational companies might display cultural differences better. However, we localized all measurements in the respec-tive national language, Spanish and German.”
Results.
No comments
Response: Thank you.
Discussions
Fine
Response: Thank you.
The conclusions
fine
Response: Thank you.
Others:
The paper has some editing issues.
Response: Thank you. We revised the manuscript.